# An early transition to magnetic supercriticality in star formation

T.-C. Ching[1], D. Li[1,2,3 ✉], C. Heiles[4], Z.-Y. Li[5], L. Qian[1], Y. L. Yue[1], J. Tang[1] & S. H. Jiao[1,2]

Magnetic fields have an important role in the evolution of interstellar medium and star formation[1,2]. As the only direct probe of interstellar field strength, credible Zeeman measurements remain sparse owing to the lack of suitable Zeeman probes, particularly for cold, molecular gas[3]. Here we report the detection of a magnetic field of $+3.8 \pm 0.3$ microgauss through the H I narrow self-absorption (HINSA)[4,5] towards L1544[6,7]—a well-studied prototypical prestellar core in an early transition between starless and protostellar phases[8–10] characterized by a high central number density[11] and a low central temperature[12]. A combined analysis of the Zeeman measurements of quasar H I absorption, H I emission, OH emission and HINSA reveals a coherent magnetic field from the atomic cold neutral medium (CNM) to the molecular envelope. The molecular envelope traced by the HINSA is found to be magnetically supercritical, with a field strength comparable to that of the surrounding diffuse, magnetically subcritical CNM despite a large increase in density. The reduction of the magnetic flux relative to the mass, which is necessary for star formation, thus seems to have already happened during the transition from the diffuse CNM to the molecular gas traced by the HINSA. This is earlier than envisioned in the classical picture where magnetically supercritical cores capable of collapsing into stars form out of magnetically subcritical envelopes[13,14].

In non-masing interstellar medium, only H I, OH and CN successfully produced systematic Zeeman measurements. The comprehensive Zeeman surveys[15] indicate that the magnetic fields in the diffuse cold neutral medium (CNM) probed by H I do not scale significantly with density, whereas above a critical break-point density of approximatey $300 \, \text{cm}^{-3}$, the magnetic fields in dense cores probed by OH tend to increase with density. However, owing to the gap in densities between the H I (about $40 \, \text{cm}^{-3}$) and OH ($\geq 10^3 \, \text{cm}^{-3}$) Zeeman measurements, the field transition around the critical density of about $300 \, \text{cm}^{-3}$ (where the dependence of the field strength on the density changes behaviour) remains a controversial topic and could have crucial implications for star formation[16–19]. Recently, a CCS Zeeman detection[20] shed light on regions denser than that probed by OH. A Zeeman probe that is sensitive for a wide range of densities, particularly the low-density molecular envelope, is highly desirable and could help to distinguish different core-formation scenarios.

We developed the so-called H I narrow self-absorption (HINSA) technique to provide a probe of the transition from H I to $H_2$ (refs. [4,5]). HINSA traces cold atomic hydrogen well mixed with $H_2$, which provides the necessary cooling, not available in the CNM, of H I through collision. Close to the steady state between $H_2$ formation and destruction, the HINSA strength is independent of the gas density[5] and thus capable of probing the transition around the critical density. Although the Zeeman effect of the H I self-absorption feature has been reported[21,22], the broad line widths of the absorption components are mostly associated with

diffuse atomic gas rather than dense molecular gas. Considering that HINSA typically has much higher brightness temperatures than most molecular lines, is impervious to depletion[23] and can be detected in a wide range of $H_2$ densities, HINSA is a promising Zeeman probe for molecular gas.

The HINSA feature in L1544 has a strong absorption dip and a nearly thermalized narrow line width at a temperature lower than 15 K (ref. [4]). The non-thermal line width and centroid velocity of the HINSA are very close to those of the emission lines of OH, $^{13}$CO and $C^{18}$O molecules, and their column densities are well correlated, suggesting that a significant fraction of the atomic hydrogen is located in the cold, well-shielded portions of L1544[5]. We thus assume that the column density sampled by the HINSA can be approximated by that obtained from dust, despite the substantially larger apparent area covered by HINSA (Fig. 1a). The previous OH Zeeman detection with Arecibo[24] towards the L1544 centre resulted in a field strength of $B_{los} = +10.8 \pm 1.7 \, \mu\text{G}$, where $B_{los}$ is the magnetic field component along the line of sight, with a positive sign representing the field pointing away from the observer. In contrast, the OH Zeeman observations of the Green Bank Telescope (GBT) towards four envelope locations 6.0′ (0.24 pc) from the centre yielded a marginal detection of $B_{los} = +2 \pm 3 \, \mu\text{G}$ (ref. [16]), leaving the structure of the envelope field undetermined.

With the Five-hundred-meter Aperture Spherical radio Telescope (FAST)[25], we detected Zeeman splittings in a 2.9′ beam (0.12 pc) towards the HINSA column density peak, 3.6′ (0.15 pc) away from the L1544

[1]National Astronomical Observatories, Chinese Academy of Sciences, Beijing, China. [2]Department of Astronomy, University of Chinese Academy of Sciences, Beijing, China. [3]NAOC-UKZN Computational Astrophysics Centre, University of KwaZulu-Natal, Durban, South Africa. [4]Department of Astronomy, University of California, Berkeley, Berkeley, CA, USA. [5]Astronomy Department, University of Virginia, Charlottesville, VA, USA. ✉e-mail: dili@nao.cas.cn

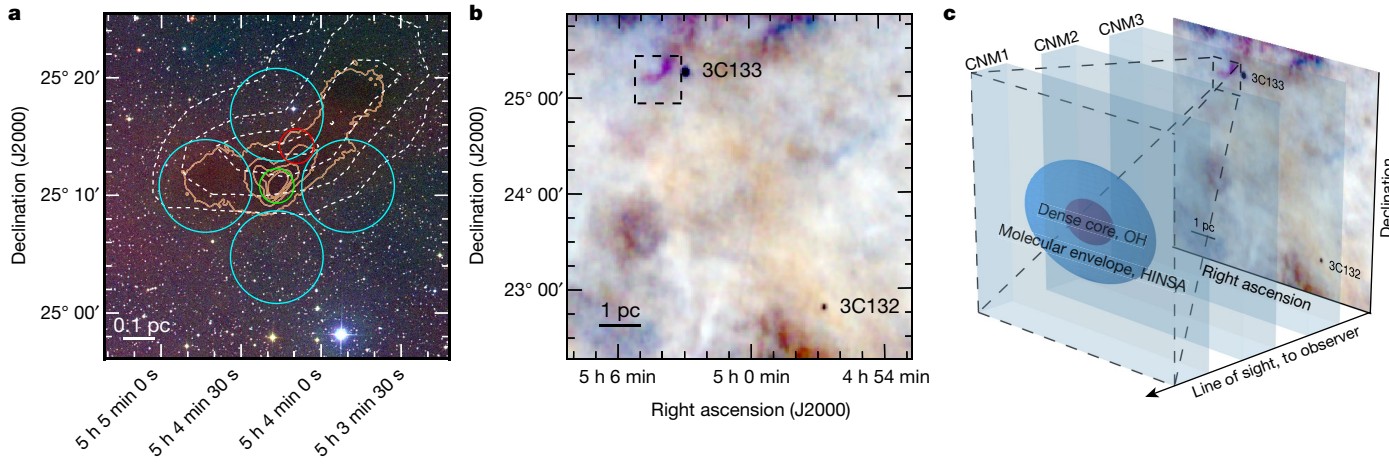

**Fig. 1 | L1544 core and illustration of the structure of interstellar medium from the CNM to the core. a**, A composite of Digitized Sky Survey 2 (DSS2) images of L1544 with the i band in red, the r band in green and the b band in blue overlaid with the HINSA and $H_2$ column density maps. The white dashed contours are 30%, 50%, 70% and 90% of the peak HINSA column density, and the orange contours are $2 \times 10^{21}$ cm$^{-2}$, $4 \times 10^{21}$ cm$^{-2}$, $6 \times 10^{21}$ cm$^{-2}$, $8 \times 10^{21}$ cm$^{-2}$ and $10 \times 10^{21}$ cm$^{-2}$ for the $H_2$ column density. The red, green and cyan circles mark the locations and beam sizes of the FAST, Arecibo and GBT Zeeman observations, respectively. **b**, A composite of three 0.5 km s$^{-1}$ velocity slices of the Arecibo GALFA-H I images at 6.2 km s$^{-1}$, 6.7 km s$^{-1}$ and 7.3 km s$^{-1}$ local standard of rest (LSR) velocities in blue, green and red. The dashed rectangle shows the region of **a**. The two absorption dots represent the locations of quasars 3C132 and 3C133. **c**, Schematic view of the CNMs, the molecular envelope and the L1544 core.

centre (Fig. 1). The spectra of the Stokes $I(v)$ and $V(v)$ parameters (where $v$ denotes velocity) are shown in Fig. 2. The $I(v)$ spectrum contains the H I emission of the CNM and the warm neutral medium (WNM) clouds in the direction towards the Taurus complex and a HINSA feature at the centroid velocity of L1544. Figure 2a shows our decomposition of $I(v)$ into a foreground HINSA component, a background WNM component, and three CNM components between the HINSA and WNM. Our fitted parameters of the HINSA component are in good agreement with the previous HINSA observations[4,5], and our parameters of the CNM and WNM components are similar to the Arecibo results towards quasars around L1544[26].

The $V(v)$ spectrum shows features of classic 'S curve' patterns proportional to the first derivatives of $I(v)$ for the HINSA, CNM and WNM components, as expected for Zeeman splittings. The Zeeman splitting profile of HINSA has a maximum at high velocity and a minimum at low velocity, opposite to the Zeeman splitting profile of CNM1, the closest CNM component at a velocity similar to L1544, which shows positive $V$ at low velocity and negative $V$ at high velocity. From our least-squares fits to $V(v)$, Fig. 2b shows the Zeeman splitting of the HINSA and the total Zeeman profile of the five components, and Fig. 3 shows the individual Zeeman splittings and $B_{los}$ of the components. The HINSA Zeeman effect gives $B_{los} = +3.8 \pm 0.3$ µG, and the H I Zeeman effect of CNM1 gives $B_{los} = +4.0 \pm 1.1$ µG. The magnetic field strengths of HINSA and CNM1 are consistent with the results of $B_{los} = +5.8 \pm 1.1$ µG and $B_{los} = +4.2 \pm 1.0$ µG obtained from the Zeeman observations towards quasars 3C133 and 3C132, probing the magnetic fields of CNM1 at distances of 17.7' (0.72 pc) and 174.5' (7.1 pc) from L1544, respectively[27]. For the second and third CNM components (CNM2 and CNM3) along the line of sight, our results of $B_{los,CNM2} = -7.6 \pm 1.0$ µG and $B_{los,CNM3} = +2.9 \pm 0.4$ µG are also consistent with the results of $B_{los,CNM2} = -9.6 \pm 6.3$ µG and $B_{los,CNM3} = -0.3 \pm 1.7$ µG towards quasar 3C133[27].

Comparing the Zeeman observations of HINSA, OH and H I tracing the CNM1 and the molecular envelope of L1544, it is clear that the magnetic fields at distances of 0.15 pc, 0.24 pc, 0.72 pc and 7.1 pc from the centre all have the same direction of $B_{los}$ and consistent strengths roughly within the 1$\sigma$. This finding is in agreement with the conclusion of a median value of 6 µG in absolute total strength in H I clouds inferred from comprehensive Zeeman surveys[15]. The HINSA Zeeman effect thus provides a connection between the magnetic fields from H I clouds to

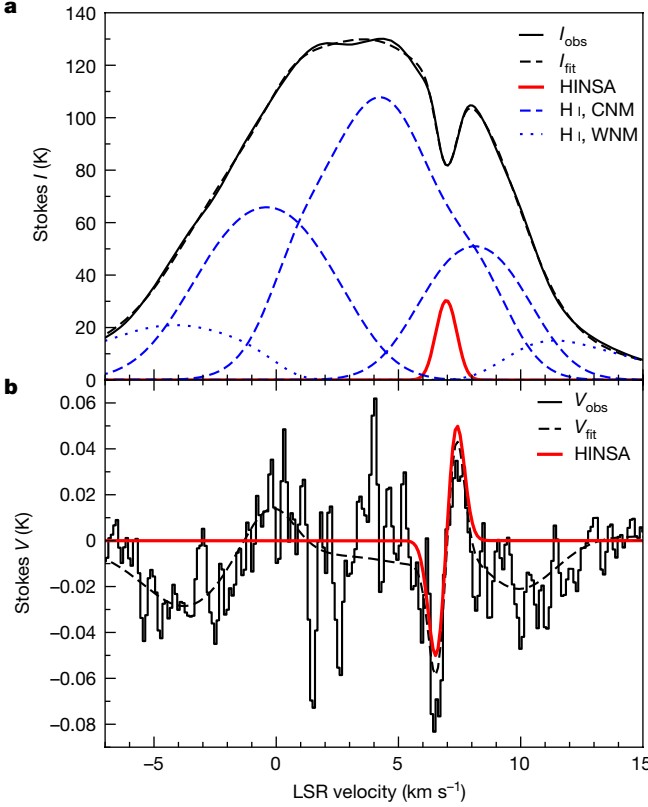

**Fig. 2 | The Stokes $I(v)$ and $V(v)$ spectra at 21-cm wavelength towards the HINSA column density peak. a**, The black profile represents the $I(v)$ spectrum. The red profile represents the absorption from the foreground HINSA component. The blue dashed and dotted profiles represent the emission of the CNM and WNM components, respectively. The CNM and WNM profiles include the absorption from the CNM components that lie in front but do not include the absorption from the HINSA. The black dashed profile represents the sum of the absorption and emission profiles. **b**, The black profile represents the $V(v)$ spectrum. The black dashed profile represents the sum of the Zeeman splitting profiles of the five components. The red profile represents the Zeeman splitting profile with $B_{los} = +3.8$ µG of the HINSA component.

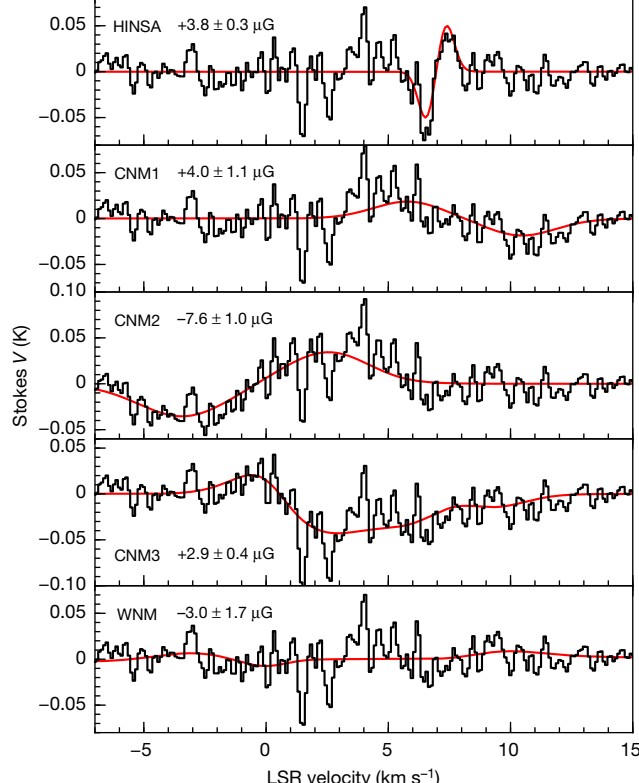

**Fig. 3 | The individual $V(v)$ profiles for the HINSA, CNM and WNM components.** In each panel, the red profile represents the fitted Zeeman profile of the component and the black profile represents the observed $V(v)$ subtracted by the fitted Zeeman profiles of the other four components. The CNM and WNM Zeeman profiles include the absorption from the CNM components that lie in front but do not include the absorption from the HINSA. The sum of the red profiles of the five components is the black dashed profile in Fig. 2b.

molecular clouds. The H I emission components (CNM1, CNM2 and CNM3) and the H I absorption towards 3C133 and 3C132 trace the CNM with a kinematic temperature of about 100 K (ref. [27]) and a number density of about 40 cm$^{-3}$ (ref. [15]), whereas the HINSA and OH observations trace the envelope of about 10–15 K (ref. [4]) and about $10^3$ cm$^{-3}$ (refs. [5,15]). Despite the one to two orders of magnitude change in both temperature and density in the phase transition from the atomic CNM to the molecular envelope, the Zeeman observations reveal a magnetic field that is coherent in both direction and strength across multi-scales and multi-phases of the interstellar medium. To constrain the uniformity of the coherent magnetic field, our likelihood analysis of the HINSA, OH and H I Zeeman measurements suggests a Gaussian distribution of the $B_{los}$ with a mean strength of $B_0 = +4.1 \pm 1.6$ µG and an intrinsic spread of $\sigma_0 = 1.2^{+1.2}_{-0.6}$ µG, a significantly better constraint than the previous estimation of $B_0 = +4^{+10}_{-8}$ µG based on only the OH results[18].

It is well known that the progenitor of molecular gas, the atomic CNM, is strongly magnetized, as measured by the dimensionless mass-to-magnetic-flux ratio $\lambda$ in units of the critical value of $2\pi G^{1/2}$ ($\lambda = 7.6 \times 10^{-21}[N_{H2}$ (cm$^{-2}$)][$B_{tot}$ (µG)]$^{-1}$, where $N_{H2}$ is the column density of $H_2$ gas and $B_{tot}$ is the total magnetic field strength[1]), which is well below unity (that is, magnetically subcritical)[28]. However, the immediate progenitors of stars, the prestellar cores of molecular clouds such as the L1554 core, are observed to be magnetically supercritical ($\lambda > 1$)[24], which is required for the self-gravity to overwhelm the magnetic support and form stars through gravitational collapse. When and how the transition from the magnetically subcritical CNM that is incapable of forming stars through direct gravitational collapse to the supercritical star-forming cores occurs is a central unresolved question in star formation.

**Table 1 | Physical parameters of the clouds**

| Tracer/cloud | $B_{los}$ (µG) | $N_{H2}$ ($10^{20}$ cm$^{-2}$) | $\lambda$[c] |
|---|---|---|---|
| H I$_{3C132}$/CNM1 | +4.2 ± 1.0 | 2.03 ± 0.54[a] | 0.18 ± 0.07 |
| H I$_{3C133}$/CNM1 | +5.8 ± 1.1 | 1.45 ± 0.52[a] | 0.10 ± 0.04 |
| HINSA/envelope | +3.8 ± 0.3 | 34.9 ± 0.1[b] | 3.5 ± 0.3 |
| OH/core | +10.8 ± 1.7 | 70.5 ± 0.2[b] | 2.5 ± 0.4 |

[a]Equivalent $H_2$ column density obtained from the H I column density listed in ref. [26].
[b]Obtained from the $H_2$ column density map in Fig. 1a. The $N_{H2}$ of the core is consistent with the value of about $9 \times 10^{21}$ cm$^{-2}$ estimated based on OH abundance[16].
[c]Obtained with $B_{tot} = 2B_{los}$.

Our HINSA Zeeman observations can be used to address this question. Using the physical parameters of the clouds (Table 1) and the statistically most probable value of $B_{tot} = 2B_{los}$, the $\lambda$ of CNM1 is about 0.10–0.18, consistent with previous results[28]. The $\lambda$ of the envelope and core of L1544 core is 2.5–3.5, which is well above unity, indicating that the transition to magnetic supercriticality has already occurred. We further consider the relative values of $\lambda$ between CNM1 and L1544 to avoid the geometrical correction from $B_{los}$ to $B_{tot}$ (ref. [16]), assuming that the inclination angles of the magnetic fields in the L1544 core and envelope are similar. Therefore, the molecular envelope of the L1544 core traced by HINSA is at least 13 times less magnetized relative to its mass compared with its ambient CNM. This is different from the 'classic' theory of low-mass star formation, which envisions the transition from magnetic subcriticality to supercriticality occurring as the supercritical core forms out of the magnetically supported (subcritical) envelope[13,14]. Our results suggest that the transition from magnetic subcriticality to supercriticality occurs earlier, during the formation of the molecular envelope, favouring the more rapidly evolving scenario of core formation and evolution for L1544[8] over the slower, magnetically retarded scenario[9]. In other words, by the time that the molecular envelope is formed, the problem of excessive magnetic flux as a fundamental obstacle to gravitational collapse and star formation is already resolved. This early reduction of flux relative to mass is unlikely owing to the 'classical' scenario where gravity drives neutrals through ions (and the magnetic field tied to them) in a process called 'ambipolar diffusion' because the CNM is not self-gravitating. The coherent magnetic fields reviewed here provide a new specific question on how to create supercritical dense cores such as L1544 from subcritical clouds. Plausible scenarios include mass accumulation along field lines[29] and (turbulence enhanced) magnetic reconnection[30], although whether such scenarios can reproduce the distributions of gas and magnetic field observed in the L1544 region remains to be seen. In any case, the already magnetically supercritical envelope can in principle go on to form dense cores and stars without having to further reduce its magnetic flux relative to the mass.

## Online content

1. McKee, C. F. & Ostriker, E. C. Theory of star formation. *Annu. Rev. Astron. Astrophys.* **45**, 565–687 (2007).
2. Hennebelle, P. & Inutsuka, S.-i. The role of magnetic field in molecular cloud formation and evolution. *Front. Astron. Space Sci.* **6**, 5 (2019).
3. Crutcher, R. M. & Kemball, A. J. Review of Zeeman effect observations of regions of star formation. *Front. Astron. Space Sci.* **6**, 66 (2019).
4. Li, D. & Goldsmith, P. F. H I narrow self-absorption in dark clouds. *Astrophys. J.* **585**, 823–839 (2003).

# Article

5.  Goldsmith, P. F. & Li, D. H I narrow self-absorption in dark clouds: correlations with molecular gas and implications for cloud evolution and star formation. *Astrophys. J.* **622**, 938–958 (2005).
6.  Elias, J. H. A study of the Taurus dark cloud complex. *Astrophys. J.* **224**, 857–872 (1978).
7.  Tafalla, M. et al. L1544: a starless dense core with extended inward motions. *Astrophys. J.* **504**, 900–914 (1998).
8.  Aikawa, Y., Ohashi, N., Inutsuka, S.-i., Herbst, E. & Takakuwa, S. Molecular evolution in collapsing prestellar cores. *Astrophys. J.* **552**, 639–653 (2001).
9.  Li, Z.-Y., Shematovich, V. I., Wiebe, D. S. & Shustov, B. M. A coupled dynamical and chemical model of starless cores of magnetized molecular clouds. I. Formulation and initial results. *Astrophys. J.* **569**, 792–802 (2002).
10. Keto, E., Caselli, P. & Rawlings, J. The dynamics of collapsing cores and star formation. *Mon. Not. R. Astron. Soc.* **446**, 3731–3740 (2015).
11. Caselli, P. et al. The central 1000 AU of a pre-stellar core revealed with ALMA. I. 1.3 mm continuum observations. *Astrophys. J.* **874**, 89 (2019).
12. Crapsi, A., Caselli, P., Walmsley, M. C. & Tafalla, M. Observing the gas temperature drop in the high-density nucleus of L 1544. *Astron. Astrophys.* **470**, 221–230 (2007).
13. Shu, F. H., Adams, F. C. & Lizano, S. Star formation in molecular clouds: observation and theory. *Annu. Rev. Astron. Astrophys.* **25**, 23–81 (1987).
14. Mouschovias, T. C. & Ciolek, G. E. in *The Origin of Stars and Planetary Systems* (eds Lada, C. J. & Kylafis, N. D.) 305–340 (NATO Advanced Study Institute Series C, **Vol. 540**, Springer, 1999).
15. Crutcher, R. M., Wandelt, B., Heiles, C., Falgarone, E. & Troland, T. H. Magnetic fields in interstellar clouds from Zeeman observations: inference of total field strengths by Bayesian analysis. *Astrophys. J.* **725**, 466–479 (2011).
16. Crutcher, R. M., Hakobian, N. & Troland, T. H. Testing magnetic star formation theory. *Astrophys. J.* **692**, 844–855 (2009).
17. Crutcher, R. M., Hakobian, N. & Troland, T. H. Self-consistent analysis of OH Zeeman observations. *Astrophys. J.* **402**, L64–L66 (2010).
18. Mouschovias, T. C. & Tassis, K. Testing molecular-cloud fragmentation theories: self-consistent analysis of OH Zeeman observations. *Mon. Not. R. Astron. Soc.* **400**, L15–L19 (2009).
19. Mouschovias, T. C. & Tassis, K. Self-consistent analysis of OH-Zeeman observations: too much noise about noise. *Mon. Not. R. Astron. Soc.* **409**, 801–807 (2010).
20. Nakamura, F. et al. First clear detection of the CCS Zeeman splitting toward the pre-stellar core, Taurus molecular cloud 1. *Publ. Astron. Soc. Jpn* **71**, psz102 (2019).
21. Goodman, A. A. & Heiles, C. The magnetic field in the Ophiuchus dark cloud complex. *Astrophys. J.* **424**, 208–221 (1994).
22. Heiles, C. A holistic view of the magnetic field in the Eridanus/Orion region. *Astrophys. J. Suppl. Ser.* **111**, 245–288 (1997).
23. Goldsmith, P. F., Li, D. & Krčo, M. The transition from atomic to molecular hydrogen in interstellar clouds: 21 cm signature of the evolution of cold atomic hydrogen in dense clouds. *Astrophys. J.* **654**, 273–289 (2007).
24. Crutcher, R. M. & Troland, T. H. OH Zeeman measurement of the magnetic field in the L1544 core. *Astrophys. J.* **537**, L139–L142 (2000).
25. Li, D. et al. FAST in space: considerations for a multibeam, multipurpose survey using China's 500-m aperture spherical radio telescope (FAST). *IEEE Microw. Mag.* **19**, 112–119 (2018).
26. Heiles, C. & Troland, T. H. The Millennium Arecibo 21 Centimeter Absorption-Line Survey. I. Techniques and Gaussian fits. *Astrophys. J. Suppl. Ser.* **145**, 329–354 (2003).
27. Heiles, C. & Troland, T. H. The Millennium Arecibo 21 Centimeter Absorption-Line Survey. III. Techniques for spectral polarization and results for Stokes *V*. *Astrophys. J. Suppl. Ser.* **151**, 271–297 (2004).
28. Heiles, C. & Troland, T. H. The Millennium Arecibo 21 Centimeter Absorption-Line Survey. IV. Statistics of magnetic field, column density, and turbulence. *Astrophys. J.* **624**, 773–793 (2005).
29. Vázquez-Semadeni, E. et al. Molecular cloud evolution—IV. Magnetic fields, ambipolar diffusion and the star formation efficiency. *Mon. Not. R. Astron. Soc.* **414**, 2511–2527 (2011).
30. Lazarian, A., Esquivel, A. & Crutcher, R. Magnetization of cloud cores and envelopes and other observational consequences of reconnection diffusion. *Astrophys. J.* **757**, 154 (2012).

# Methods

## Data reduction

The FAST Zeeman observations towards the HINSA column density peak in L1544 were carried out on five days between August and November 2019 with a total integration time of 7.6 h. The HINSA spectra were obtained with the central beam of the L-band 19-beam receiver[31]. The central beam has an average system temperature of 24 K, a main beam efficiency of 0.63 and a main beam diameter at the half-power point of 2.9′ with a pointing accuracy of 7.9″. The 19-beam receiver had orthogonal linear polarization feeds followed by a temperature-stabilized noise injection system and low noise amplifiers to produce the X and Y signals of the two polarization paths. The XX, YY, XY and YX correlations of the signals then were simultaneously recorded using the ROACH backend with 65,536 spectral channels in each polarization. The spectral bandwidth was 32.75 MHz centred at the frequency of the H I 21-cm line for a channel spacing of 500 Hz, and the $V(v)$ spectrum presented in this work was Hanning smoothed, which produced a spectral resolution of 0.21 km s$^{-1}$.

The data reduction, including gain and phase calibrations of the two polarization paths, bandpass calibrations of the four correlated spectra and polarization calibrations to generate the Stokes $I$, $Q$, $U$ and $V$ spectra, was carried out using the IDL RHSTK package written by C. Heiles and T. Robishaw, which is widely used for Arecibo and GBT polarization data. The 19-beam receiver is rotatable from −80° to +80° with respect to the line of equatorial latitude. The polarization calibrations used drifting scans of the continuum source 3C286 at rotation angles of −60°, −30°, 0°, 30° and 60° over 1.5 h surrounding its transit. The details of the polarization calibration procedure are provided in ref. [32]. We performed polarization calibrations once per month during the observations. The calibrated polarization of 3C286 of the three epochs were 8.9% ± 0.1%, 8.7% ± 0.2% and 9.0% ± 0.1% for polarization degrees and 30.4° ± 0.3°, 33.8° ± 0.5° and 29.4° ± 0.3° for polarization angles. Considering that the ionosphere can generate a Faraday rotation of 1°–3° in polarization angle at the L band[33], our results were consistent with the intrinsic polarization degree of 9.5% and polarization angle of 33° of 3C286 at 1,450 MHz (ref. [34]). In addition to the polarization observations of L1544 and 3C286, we observed the circularly polarized OH maser source IRAS02524+2046[35] to verify that our procedures produced consistent $B_{los}$, including the sign or direction of the magnetic field, as had been obtained previously.

The convolutions of the sidelobes of the Stokes $V$ beam with the spatial gradient of the Stokes $I$ emission may generate a false 'S curve' in the $V$ spectrum[27]. To check the credibility of our Zeeman detections, we measured the Stokes $V$ beam of FAST and convolved the beam with the Galactic Arecibo L-band Feed Array (GALFA) Stokes $I$ cube[36] of L1544. The convolved $V$ spectrum showed a profile with a shape similar to the $I$ spectrum and a strength less than 0.03% of the $I$ spectrum, different from the 'S curve' patterns in the observed $V$ spectrum. Meanwhile, the 19-beam receiver was rotated to −45°, 0° and 45° in the three epochs of the L1544 observations, and all of the three epochs showed 'S curve' patterns in the $V$ spectra, indicating that our Zeeman results were true detections.

Although the data of the 19 beams of the FAST L-band receiver were simultaneously taken in our observations, only the polarization of the central beam was commissioned at the time of writing. The results represented in this work were made with only the central beam pointing towards the HINSA column density peak in Fig. 1. The Zeeman results of the 18 off-central beams will be published in the future.

## Multiple Gaussians and radiative transfer fitting to $I(v)$ and $V(v)$

We adopt the least-squares fits of multiple Gaussians with radiative transfer[26] to decompose the $I(v)$ into the HINSA, CNM and WNM components. The expected profile of $I(v)$ consists of multiple CNM components providing opacity and also brightness temperature and a WNM component providing only brightness temperature:

$$I(v) = I_{CNM}(v) + I_{WNM}(v). \tag{1}$$

The $I_{CNM}(v)$ is an assembly of $N$ CNM components

$$I_{CNM}(v) = \sum_{n=1}^{N} I_{peak,n}(1 - e^{-\tau_n(v)})e^{-\left(\sum_{m=0}^{M}\tau_m(v) + \tau_0\right)}, \tag{2}$$

where the subscript $m$ with its associated optical depth profile $\tau_m(v)$ represents each of the $M$ CNM clouds that lie in front of cloud $n$. The optical depth of the $i$th component is

$$\tau_i(v) = \tau_i e^{-[(v-v_{0,i})/\sigma_{v,i}]^2} \tag{3}$$

in which $\tau_0$ represents the HINSA providing only opacity and no brightness temperature. For the WNM in the background

$$I_{WNM}(v) = I_{peak,WNM}e^{-[(v-v_{0,WNM})/\sigma_{v,WNM}]^2}e^{-\sum_{i=0}^{N}\tau_i(v)}. \tag{4}$$

The fitting of $I(v)$ thus yields values for the intrinsic peak Stokes $I$ emission ($I_{peak}$), $\tau$, $v_0$ and the Gaussian dispersion ($\sigma_v$) of the components.

We consider the radiative transfer of $V(v)$ in terms of right circular polarization (RCP) and left circular polarization (LCP). The Zeeman effect states that with the existence of $B_{los}$, the frequency of RCP shifts from its original frequency $v_0$ to $v_0 + v_z$ and the frequency of LCP shifts to $v_0 - v_z$ with $v_z = (Z/2) \times B_{los}$, where $Z$ is the Zeeman splitting factor (2.8 Hz µG$^{-1}$ for the H I 21-cm line). As the RCP and LCP are orthogonal components of radiation, the radiative transfer processes of RCP and LCP are independent of each other. For RCP, equation (1) becomes

$$T_{RCP} = T_{RCP,CNM}(v, \tau_{RCP,i}) + T_{RCP,WNM}(v, \tau_{RCP,i}), \tag{5}$$

where for the $i$th component, $T_{RCP,i} = I_{peak,i}/2$, $\tau_{RCP,i}$ is the optical depth in the RCP radiation to substitute the $\tau_i$ in equation (3) with $\tau_{RCP,i} = \tau_i(v_0 + v_{z,i})$ for $B_{los,i}$ of the component, and the parameters of $v_0$ and $\sigma_v$ remain the same. Similarly, for LCP

$$T_{LCP} = T_{LCP,CNM}(v, \tau_{LCP,i}) + T_{LCP,WNM}(v, \tau_{LCP,i}) \tag{6}$$

with $T_{LCP,i} = I_{peak,i}/2$ and $\tau_{LCP,i} = \tau_i(v_0 - v_{z,i})$. The fitting of $V(v) = T_{RCP} - T_{LCP} + cI(v)$, which includes a $c$ term accounting for leakage of $I(v)$ into $V(v)$, thus yields values for $B_{los}$ of the components. In Extended Data Table 1, we list the parameters of the components obtained from least-squares fits to $I(v)$ and $V(v)$. The leakage of our HINSA Zeeman observations is $c = 0.034\%$.

## HINSA and H$_2$ column density maps

L1544 is a low-mass prestellar core in the Taurus molecular cloud complex at a distance of about 140 pc. The core has a size of about 0.1 pc (ref. [37]), presumably formed out of a parsec-long elongated molecular ridge[7], which, for simplicity, we refer as the molecular envelope. We show the HINSA and H$_2$ column density maps of L1544 in Fig. 1a, and we use the H$_2$ column density map to calculate the $N_{H2}$ of the envelope and core at the beams of FAST and Arecibo observations in Table 1. The HINSA column density map is a revision of Fig. 8 in ref. [4]. To derive the H$_2$ column density map, we retrieved the level-2.5 processed, archival Herschel images that were taken at 250 µm/350 µm/500 µm using the SPIRE instrument[38] (observation ID 1342204842). We smoothed the Herschel images to a common angular resolution of the 36″ beam at 500 µm and regridded the images to the same pixel size of 6″. We performed least-squares fits of the 250 µm/350 µm/500 µm spectral energy distributions weighted by the squares of the measured noise levels to derive the pixel-to-pixel distributions of dust temperature $T_d$ and dust

optical depth $\tau_\nu$ using $S_\nu = \Omega_m B_\mu(T_d)(1 - e^{-\tau_\nu})$, where $S_\nu$ is the flux density at frequency $\nu$, $\Omega_m$ is the is the solid angle of the pixel, $B_\mu(T_d)$ is the Planck function at $T_d$ and $\tau_\nu = \tau_{230}(\nu(\text{GHz})/230)^\beta$ with a dust opacity index $\beta$ of 1.8. Next, we obtained the $H_2$ column density with $N_{H2} = g\tau_{230}/(\kappa_{230}\mu_m m_H)$, where $g = 100$ is the gas-to-dust mass ratio, $\kappa_{230} = 0.09 \text{ cm}^2\text{ g}^{-2}$ (ref. [39]) is the dust opacity at 230 GHz, $\mu_m = 2.8$ is the mean molecular weight and $m_H$ is the atomic mass of hydrogen. To estimate the uncertainties in the $H_2$ column density, we used a Monte Carlo technique. For each pixel, we created artificial 250 μm/350 μm/500 μm flux densities by adding the original flux densities with normal-distributed errors taking account the uncertainty in the measured flux and a 10% correlation for the calibration uncertainty in SPIRE[40]. We then estimated the uncertainty in each pixel with 1,000 fittings of the $H_2$ column density. The $N_{H2}$ and its uncertainty in Table 1 were obtained from the convolutions of the $H_2$ column density map and uncertainty map with the FAST and Arecibo beams.

Note that the equivalent $H_2$ column density $N_{H2}$ of the CNM1 are derived from H I data towards 3C132 and 3C133, a method different to the $N_{H2}$ of the L1544 envelope and core that are derived from dust emission. Therefore, in addition to the statistical errors listed in Table 1, there is a systematic difference between the $N_{H2}$ derived from the two methods. Considering that the regime traced by dust emission can be different from those traced by HINSA or OH, which is particularly noticeable from the different spatial extents of dust and HINSA in Fig. 1a, we expect that the systematic difference could be as large as a factor of a few. As the values of $\lambda$ between CNM1 and L1544 are different at least by a factor of 13, the systematic difference between the two methods should not change the qualitative conclusion of this work.

In Fig. 1a, the peak of HINSA column density appears to be shifted from the centre of L1544 by 0.15 pc and the 70% and 90% contours of the peak HINSA column density do not enclose L1544. We note that such offset has also been seen for other dense gas tracers in prestellar cores[41]. The core geometry may not be as simple as envisioned in idealized theories, where the dense core sits near the centre of a lower-density molecular envelope. In particular, the L1544 core appears to sit near one end of an elongated molecular (and dust) ridge, which roughly coincides with the region traced by the HINSA. Such an offset can result from complexities in chemistry and formation history, but does not affect the main science result of this work, namely, the HINSA Zeeman probes the magnetic fields of the current molecular ridge that is the progenitor of the dense core.

## Maximum likelihood

We adopt the analysis of maximum likelihood[18] to study the uniformity of magnetic fields in the envelope of L1544. Assuming that the true $B_{los}$ follows a Gaussian distribution with mean $B_0$ and intrinsic spread $\sigma_0$, the likelihood $l_j$ for a single observation in a set of $N$ measurements ($j = 1, ..., N$) to measure $B_j$ with Gaussian error $\sigma_j$ is proportional to the convolution of the probability $\exp[-(B - B_0)^2/2\sigma_0^2]/\sqrt{2\pi}\,\sigma_0$ for the magnetic field to have a true value of $B$ with the probability $\exp[-(B - B_j)^2/2\sigma_j^2]/\sqrt{2\pi}\,\sigma_j$ of observing a value $B_j$ of the field. Therefore, $l_j$ is the integral over all possible true values of the magnetic field

$$l_j = \int_{-\infty}^{\infty} dB \frac{\exp[-(B - B_j)^2/2\sigma_j^2]}{\sqrt{2\pi}\,\sigma_j} \frac{\exp[-(B - B_0)^2/2\sigma_0^2]}{\sqrt{2\pi}\,\sigma_0}. \tag{7}$$

Although the overall likelihood $\mathcal{L}$ for a set of observations is the product of individual likelihoods of the observations $\left(\mathcal{L} = \prod_{j=1}^{N} l_j\right)$, the $B_0$ and $\sigma_0$ can be estimated by maximizing the likelihood $\mathcal{L}$. After performing the integration in equation (7) and some algebraic manipulations

$$\mathcal{L}(B_0, \sigma_0) = \left(\prod_{J=1}^{N} \frac{1}{\sqrt{\sigma_0^2 + \sigma_j^2}}\right) \exp\left[-\frac{1}{2}\sum_{j=1}^{N} \frac{(B_j - B_0)^2}{\sigma_0^2 + \sigma_j^2}\right] \tag{8}$$

Extended Data Fig. 1 shows the distribution of $\mathcal{L}$ as functions of $B_0$ and $\sigma_0$ and the probability distributions of $B_0$ and $\sigma_0$ by integrating $\mathcal{L}$ along the $B_0$ axis and the $\sigma_0$ axis, respectively. The probability distribution of $B_0$ is similar to a normal distribution with a mean value of +4.1 μG and a standard deviation of 1.6 μG. The probability distribution of $\sigma_0$ is highly asymmetric as the values of $\sigma_0$ cannot be negative. The first, second and third quartiles of the $\sigma_0$ distribution are 0.6 μG, 1.2 μG and 2.4 μG. We therefore suspect that the Zeeman measurements in the L1544 envelope can be explained by a magnetic field with $B_0 = +4.1 \pm 1.6$ μG and $s_0 = 1.2^{+1.2}_{-0.6}$ μG.

## Inclination angle of magnetic field

Given the uniformity of magnetic fields in the envelope of L1544 and CNM1 is well constrained by the maximum likelihood analysis, the coherent $B_{los}$ suggests that the inclination angles of magnetic fields in the CNM1 and L1544 envelope are likely to be similar, or a special geometry of magnetic field structure across multi-scales and multi-phases of the interstellar medium is needed. In contrast, the $B_{los}$ of the L1544 envelope and core differ by a factor of 2.6. There are two physical explanations for the 2.6-times difference between the HINSA and OH Zeeman measurements. First, the OH measurement probably samples a denser gas than the HINSA measurement, as the column density along the OH sightline is twice that along the HINSA sightline (Table 1). As the magnetic field strength in molecular clouds tends to increase with number density[15], the stronger field is naturally expected in the denser core. Alternatively, the inclination angle of the L1544 core magnetic field could differ substantially from that of the coherent field. As we cannot rule out the second possibility, an assumption of similar inclination angles of the magnetic fields in the L1544 envelope and core thus is required to calculate the relative values of $\lambda$.

We note that dust polarization observation may give some clues as dust polarization traces the position angle of the plane-of-sky component of the magnetic field. The near-infrared polarization observations of L1544[42] indicate that the mean position angle of the magnetic field towards the core location of the Arecibo beam is 29.0°–36.9°, and the mean position angles of magnetic fields towards the four envelope locations of the GBT beams are 30.5°–55.8°. The difference in the position angles between the core and envelope thus may be about 10°–20°.

We perform Monte Carlo simulations[43] to study whether the 2.6-times difference between the $B_{los}$ of the L1544 envelope and core can be explained by different inclination angles. The simulations randomly generate two unit vectors in three dimensions, and then measure the difference between the inclination angles and the difference between the position angles of the two vectors. The probability of the cases that the line-of-sight length of one vector is 2.6-times larger than that of the other is roughly 0.19. For those cases, the mean difference between the inclination angles and the mean difference between the position angles of the two vectors are about 38° and 45°, respectively. As the probability of 0.19 is small and the difference of about 45° between the simulated position angles is about a factor of two to four times larger than the difference of about 10°–20° between the observed position angles, it is less likely that the 2.6-times difference between the $B_{los}$ of the L1544 envelope and core can be solely explained by different inclination angles.

## CCS Zeeman measurements

Ref. [20] reported a CCS Zeeman detection of $117 \pm 21$ μG in the dense core of TMC-1 that has an estimated $H_2$ column density of $3 \times 10^{22}$ cm$^{-2}$, which is four-times higher than that probed by OH Zeeman measurements in L1544 and nearly one order of magnitude higher than that probed by our HINSA measurements. It appears to provide further support to the evolutionary scenario suggested by our HINSA measurements: namely, once the gas loses its magnetic support during the transition from the CNM to the molecular envelope (or ridge) and becomes magnetically supercritical, there is no longer any need to lose magnetic flux further

(relative to the mass) for a piece of the envelope/ridge to condense into a (magnetically supercritical) core (for example, the L1544 core probed by OH) and for the core to evolve further by increasing its column density (for example, the TMC-1 core probed by CCS).

Technically, we note that one potential source of significant uncertainty in frequency shift, namely the uncertainty of beam squint, was not included in the CCS result, which may affect the level of significance. In comparison, the HINSA measurement is robust with a greater than $10\sigma$ significance with the beam squint and velocity gradient being taken into account by convolving the FAST Stokes $V$ beam with the Stokes $I$ cube of L1544 (see the third paragraph in the data reduction section in Methods).

## Data availability

The data that support the findings of this study are openly available in Science Data Bank at https://www.doi.org/10.11922/sciencedb.01221. The FAST raw data are available from the http://fast.bao.ac.cn site one year after data-taking, per the data policy of FAST. Owing to the large data volume of this work and the speciality of polarization calibration, interested users are encouraged to contact the corresponding authors to arrange data transfer. The reduced $I(v)$ and $V(v)$ spectra are available at https://github.com/taochung/HINSAzeeman.

## Code availability

The codes analysing the $I(v)$ and $V(v)$ spectra reported here are available at https://github.com/taochung/HINSAzeeman. The IDL RHSTK package is available at http://w.astro.berkeley.edu/heiles/.

31. Jiang, P. et al. The fundamental performance of FAST with 19-beam receiver at L band. *Res. Astron. Astrophys.* **20**, 064 (2020).
32. Heiles, C. et al. Mueller matrix parameters for radio telescopes and their observational determination. *Publ. Astron. Soc. Pac.* **113**, 1274–1288 (2001).
33. Jehle, M., Ruegg, M., Zuberbuhler, L., Small, D. & Meier, E. Measurement of ionospheric Faraday rotation in simulated and real spaceborne SAR data. *IEEE Trans. Geosci. Remote Sens.* **47**, 1512–1523 (2009).
34. Perley, R. A. & Butler, B. J. Integrated polarization properties of 3C48, 3C138, 3C147, and 3C286. *Astrophys. J. Suppl. Ser.* **206**, 16 (2013).
35. McBride, J. & Heiles, C. An Arecibo survey for Zeeman Sslitting in OH megamaser galaxies. *Astrophys. J.* **763**, 8 (2013).
36. Peek, J. E. G. et al. The GALFA-H I Survey data release 2. *Astrophys. J. Suppl. Ser.* **234**, 2 (2018).
37. Ward-Thompson, D., Motte, F. & Andre, P. The initial conditions of isolated star formation—III. Millimetre continuum mapping of pre-stellar cores. *Mon. Not. R. Astron. Soc.* **305**, 143–150 (1999).
38. Griffin, M. J. et al. The Herschel-SPIRE instrument and its in-flight performance. *Astron. Astrophys.* **518**, L3 (2010).
39. Ossenkopf, V. & Henning, T. Dust opacities for protostellar cores. *Astron. Astrophys.* **291**, 943–959 (1994).
40. Roy, A. et al. Reconstructing the density and temperature structure of prestellar cores from Herschel data: a case study for B68 and L1689B. *Astron. Astrophys.* **562**, A138 (2014).
41. Lai, S.-P., Velusamy, T., Langer, W. D. & Kuiper, T. B. H. The physical and chemical status of pre-protostellar core B68. *Astron. J.* **126**, 311–318 (2003).
42. Clemens, D. P., Tassis, K. & Goldsmith, P. F. The magnetic field of L1544. I. Near-infrared polarimetry and the non-uniform envelope. *Astrophys. J.* **833**, 176 (2016).
43. Hull, C. L. H. et al. Misalignment of magnetic fields and outflows in protostellar cores. *Astrophys. J.* **768**, 159 (2013).

**Acknowledgements** This work was supported by the National Natural Science Foundation of China (NSFC) grant numbers 11988101, U1931117 and 11725313; by the CAS International Partnership Program number 114A11KYSB20160008; by the National Key R&D Program of China number 2017YFA0402600; and by the Cultivation Project for FAST Scientific Payoff and Research Achievement of CAMS-CAS. T.-C.C. is funded by the Chinese Academy of Sciences Taiwan Young Talent Program grant number 2018TW2JB0002. T.-C.C. and J.T. were supported by Special Funding for Advanced Users, budgeted and administrated by Center for Astronomical Mega-Science (CAMS), Chinese Academy of Sciences. C.H. is funded by the Chinese Academy of Sciences President's International Fellowship Initiative grant number 2020DM0005. Z.-Y.L. is supported in part by NASA 80NSSC20K0533 and NSF AST-1716259 and 1815784. This work made use of data from FAST, a Chinese national mega-science facility built and operated by the National Astronomical Observatories, Chinese Academy of Sciences. This research has made use of the services of the DSS2 survey of the ESO Science Archive Facility.

**Author contributions** T.-C.C., D.L. and C.H. launched the FAST Zeeman project; T.-C.C. processed the data and analysis in consultation with C.H.; T.-C.C., Z.-Y.L., D.L. and C.H. drafted the paper; L.Q., Y.L.Y. and J.T. made key contributions to arrange the FAST observations of L1544 and polarization calibration; S.H.J. provided the $H_2$ column density map.

**Competing interests** The authors declare no competing interests.

**Additional information**
**Correspondence and requests for materials** should be addressed to D. Li.

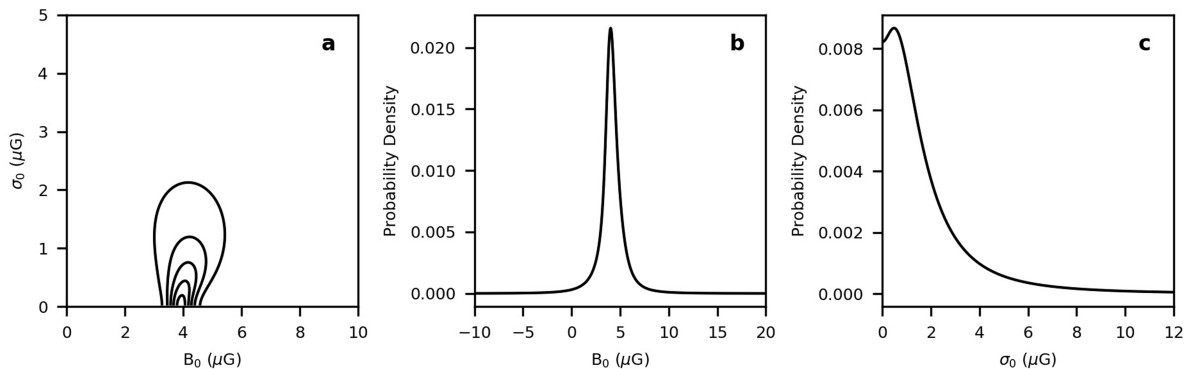

**Extended Data Fig. 1 | Likelihood** $\mathcal{L}$ **for the coherent magnetic field to have mean ($B_0$) and spread ($\sigma_0$) values. a**, Contours of $\mathcal{L}$ as functions of $B_0$ and $\sigma_0$ plotted at 10%, 30%, 50%, 70% and 90% of the peak value. **b**, The probability distribution of $B_0$ while allowing all possible values of $\sigma_0$. **c**, The probability distribution of $\sigma_0$ while allowing all possible values of $B_0$.

**Extended Data Table 1 | Gaussian fit parameters**

| Component | $I_{peak}$ [K]* | $\tau$ † | $v_{LSR}$ [km s$^{-1}$] ‡ | $\sigma_v$ [km s$^{-1}$] § | $B_{los}$ [$\mu$G] | Order ‖ |
|---|---|---|---|---|---|---|
| HiNSA | – | 0.32 ± 0.01 | 6.97 ± 0.01 | 0.40 ± 0.01 | +3.8 ± 0.3 | 0 |
| CNM1 | 90.34 ± 5.49 | 0.83 ± 0.12 | 8.12 ± 0.11 | 1.86 ± 0.05 | +4.0 ± 1.1 | 1 |
| CNM2 | 116.33 ± 1.78 | 0.84 ± 0.08 | -0.39 ± 0.33 | 2.41 ± 0.13 | -7.6 ± 1.0 | 2 |
| CNM3 | 135.31 ± 2.04 | 10.45 ± 0.95 | 4.38 ± 0.09 | 2.04 ± 0.06 | +2.9 ± 0.4 | 3 |
| WNM | 46.70 ± 2.47 | – | 2.63 ± 0.04 | 6.44 ± 0.09 | -3.0 ± 1.7 | 4 |

*$I_{peak}$ is the intrinsic peak Stokes $I$ emission. We do not fit $I_{peak}$ for the HINSA because it is an absorption component.
†$\tau$ is the central opacity. We do not fit $\tau$ for the WNM because it is a background component.
‡$v_{LSR}$ is the central LSR velocity.
§$\sigma_v$ is the Gaussian dispersion.
‖The order of the component along the line of sight. Order begins with 0, and increasing numbers mean increasing distance along the line of sight. We fix order to 0 for the HINSA as the Taurus cloud is one of the closest cloud to us. The orders of the other components are free parameters in the fitting.