## [Peer Review File · Nature]

Manuscript Title: An Early Transition to Magnetic Supercriticality in Star Formation

Editorial Notes:

Reviewer Comments & Author Rebuttals

Reviewer Reports on the Initial Version:

Referee #1 (Remarks to the Author):

This paper presents sensitive H I Zeeman observations toward the L1544 molecular cloud. The simultaneous fitting of five spectral-line components for the magnetic field strengths is an impressive tour de force. The result is the first detection of the Zeeman effect in cold H I self-absorption. This is an extremely important observational result as it provides a new tool for measuring magnetic field strengths in the interstellar medium and in particular in the envelopes of the dense molecular clouds where star formation takes place. There remains considerable uncertainty in the role that magnetic fields play in the star formation process, so the addition of this tool is important. A crucial parameter λ in understanding the evolution of molecular clouds is the ratio of mass to magnetic flux with respect to the critical value for magnetic pressure support against gravity. The authors analyze their result together with published Zeeman results for the surrounding warmer H I and the molecular core sampled by OH to report that λ changes from subcritical (support) to supercritical (non support) in the molecular envelope of L1544, which they state is an earlier evolutionary stage than called for by a "classical" star formation theory. The result would therefore be extremely important in informing our understanding of star formation.

Although I will strongly recommend that the paper be published for the above reasons, there are several issues that must first be addressed. The central new astrophysics in this paper is that λ goes from subcritical in the extended cold neutral medium near L1544 to supercritical in the molecular envelope rather than just in the dense core. This assertion comes from table 1, where the field strength is the same in the first three rows but the column density N increases by an order of magnitude from rows 1 and 2 to row 3. One must therefore look at these numbers carefully. The origin of N for rows 1 and 2 is reference 27 (not 26 as stated). These are H I column densities, inferred from the H I lines. The row 3 N is by a completely different method, raising the possibility of a systematic difference. The actual H I column density from the self-absorption is much, much less than what they list in the table. The N they list comes from the assertion that the H I self-absorption samples the molecular envelope of the cloud, and that therefore the dust emission may be used to infer the column density that the H I self-absorption line samples. The evidence that is discussed for this (last paragraph of page 2) is subject to greater uncertainty than the paper suggests. The close agreement of velocities and line widths for the H I self-absorption (width corrected for thermal broadening), OH, and CO support the argument (as discussed in their ref. 5). They state that the column density maps are correlated. However, the extent of the H I shown in figure 1a is perhaps 3 times larger than the dust extent at the same fractional column density, suggesting that the H I may be sampling a somewhat different regime from the dust emission, and that perhaps the column density sampled by the H I differs from that of the molecular envelope. I suggest that they explicitly state that the spatial extents differ by a factor of about 3, and after the last sentence of this paragraph they add the sentence: "Based on the close agreement between the H I self-absorption, CO, and OH velocities and line widths, we assume that the column density sampled by the H I self-absorption is equal to that obtained from the dust emission in spite of the different spatial extents." The crucial word is assume. Finally, the column densities N for the H I self-absorption and OH positions in table 1 are obtained from the H_2 column density map in figure 1a. Are these H or H_2 column densities? Their reference 5 lists $N(H_2)$ inferred from CO that agrees with the N in table 1 of this paper, further suggesting that the N are $N(H_2)$. If these are truly H_2 column densities, the N 's should be doubled to yield the H

column density for use in calculating λ . That would produce better agreement with the column density inferred from OH (ref. 18) for row 4. That change would double the last two values of λ in table 1. This change would not change the astrophysical conclusion of this paper. If this confusion of $N(\text{H})$ with $N(\text{H}_2)$ is not true, they should correct the caption of figure 1a and the footnote to table 1 to state that these are $N(\text{H})$ and not $N(\text{H}_2)$, and perhaps comment on the disagreement with the ref. 5 and 18 results for row 4 of the table.

They state that their result shows that the reduction in magnetic flux relative to mass occurs earlier than envisioned in the "classical" theory of star formation, but the discussion of this is rather limited (lines 141-149). Yet this is THE astrophysical result of the paper. They should briefly describe the "classical" prediction, namely that mass/flux is reduced by gravity driving neutrals through the ions and magnetic field by a process called ambipolar diffusion. They should explicitly state exactly why their result is contrary to this prediction; that is, what is the argument that the regime sampled by the H I self-absorption is not gravitationally contracting with ambipolar diffusion producing a subcritical region as in the "classical" theory. Further, they should briefly suggest how their result might be explained theoretically if ambipolar diffusion does not. Two major possibilities are (1) formation of molecular clouds by flows along flux tubes (i.e., Vazquez-Semadeni et al., MNRAS 414, 2511, 2011) and (2) magnetic reconnection (i.e., Lazarian et al., ApJ 757, 154, 2012). In (1), for a relatively small distance along a flux tube, as sampled by a small telescope beam, initially there will be little mass and λ will be measured to be highly subcritical. As mass flows into the region of the cloud (not due to gravity but due to colliding interstellar flows), λ (again as sampled in a small telescope beam) increases. Thus clouds start out as atomic and subcritical and accumulate mass over large distances to become molecular and supercritical as they evolve, becoming self-gravitating at about the same time. In (2), as the amplitude of turbulence as well as the scale of turbulent motions decrease from the envelope to the core of a cloud, the diffusion of the magnetic field is faster in the envelope. As a result, the magnetic flux trapped during the collapse in the envelope is being released faster than the flux trapped in the core, resulting in much weaker fields in envelopes than in cores. Both of these "non-classical" theories would seem capable of explaining the observational result of this paper.

The statement in lines 138-140 that "coherence" of the magnetic fields means the angles between the fields in the envelope and core are the same should be expanded. It will be subject to the same criticism made in reference 22 to the OH Zeeman results of reference 19, that the angles between the line of sight and fields in the two regions may be different.

In table 2 the authors state the order of the five components along the line of sight, without explaining how they arrived at this order. Of course the self-absorption must be closer than at least one emission component, but how was the specific order determined? In particular, would a different order affect their fitting results?

These additions may lead to the paper being too long. If so, I would recommend that the detailed discussion of maximum likelihood and figure 4 be deleted. The statement in lines 106-111 is adequate, and the above points are more important than the detailed discussion.

There are a few more minor points:

(1) In line 58 they state that "...H I ... is less affected by depletion at high densities than heavier molecules...". They should provide a reference for this assertion.

(2) Just to have the paper be very clear, I recommend that they state explicitly the + sign for the magnetic fields rather than just have the reader assume that no sign means +. The direction of the field is crucial in their argument of a "coherence" to the magnetic field in this region.

(3) In the sentence beginning on line 86, they assert that the weak field strengths are broadly consistent with the H-band polarization angles. This assertion is so broad and nebulous that I

recommend the sentence be removed. Or, I know that Clemens has inferred field strengths from his H-band data to which the authors refer. The authors could ask Clemens for permission to quote his as yet unpublished results if they feel this point is important.

(4) Whenever B is used to mean the total field strength, I recommend adding the subscript "total" to B for clarity; and again for better clarity, in table 1, add a footnote to λ stating that $B_{\text{total}} = 2B_{\text{los}}$ is used in the calculation.

(5) The authors refer to the fact that the data were obtained with a 19-beam system, but it is unclear what if any use was made of results from other than the central beam. Were they used to produce the H I self-absorption map of fig. 1a or was only the central beam used with multiple pointings? What about Zeeman results for the other 18 beams to produce a magnetic field map?

The above points are addressable and correctable and should not prohibit publication.

These results are of immediate interest to both observers and theorists in the area of physics of the interstellar medium and star formation. The approach of this paper is generally valid, quality of the data is excellent, and quality of presentation is satisfactory (once above points have been addressed). Full details of data and methodology are presented. With regard to statistics and treatment of uncertainties, they authors should note that the uncertainties listed in table 1 are statistical only and do not include systematic uncertainties, especially in the column density. With the suggested explicit statement of the assumption, the conclusions and data interpretation are robust, valid and reliable. The manuscript does reference previous literature appropriately, once discussion and reference to the other theories discussed above are added. The abstract, introduction, and conclusions are clear, accessible, and appropriate.

Author Rebuttals to Initial Comments:

Referee's Comments:

(1) *Although I will strongly recommend that the paper be published for the above reasons, there are several issues that must first be addressed. The central new astrophysics in this paper is that λ goes from subcritical in the extended cold neutral medium near L1544 to supercritical in the molecular envelope rather than just in the dense core. This assertion comes from table 1, where the field strength is the same in the first three rows but the column density N increases by an order of magnitude from rows 1 and 2 to row 3. One must therefore look at these numbers carefully. The origin of N for rows 1 and 2 is reference 27 (not 26 as stated). These are H_I column densities, inferred from the H_I lines. The row 3 N is by a completely different method, raising the possibility of a systematic difference. The actual H_I column density from the self-absorption is much, much less than what they list in the table. The N they list comes from the assertion that the H_I self-absorption samples the molecular envelope of the cloud, and that therefore the dust emission may be used to infer the column density that the H_I self-absorption line samples. The evidence that is discussed for this (last paragraph of page 2) is subject to greater uncertainty than the paper suggests. The close agreement of velocities and line widths for the H_I self-absorption (width corrected for thermal broadening), OH, and CO support the argument (as discussed in their ref. 5). They state that the column density maps are correlated. However, the extent of the H_I shown in figure 1a is perhaps 3 times larger than the dust extent at the same fractional column density, suggesting that the H_I may be sampling a somewhat different regime from the dust emission, and that perhaps the column density sampled by the H_I differs from that of the molecular envelope. I suggest that they explicitly state that the spatial extents differ by a factor of about 3,*

and after the last sentence of this paragraph they add the sentence: “Based on the close agreement between the H_I self-absorption, CO, and OH velocities and line widths, we assume that the column density sampled by the H_I self-absorption is equal to that obtained from the dust emission in spite of the different spatial extents.” The crucial word is assume. Finally, the column densities N for the H_I self-absorption and OH positions in table 1 are obtained from the H₂ column density map in figure 1a. Are these H or H₂ column densities? Their reference 5 lists $N(\text{H}_2)$ inferred from CO that agrees with the N in table 1 of this paper, further suggesting that the N are $N(\text{H}_2)$. If these are truly H₂ column densities, the N 's should be doubled to yield the H column density for use in calculating λ . That would produce better agreement with the column density inferred from OH (ref. 18) for row 4. That change would double the last two values of λ in table 1. This change would not change the astrophysical conclusion of this paper. If this confusion of $N(\text{H})$ with $N(\text{H}_2)$ is not true, they should correct the caption of figure 1a and the footnote to table 1 to state that these are $N(\text{H})$ and not $N(\text{H}_2)$, and perhaps comment on the disagreement with the ref. 5 and 18 results for row 4 of the table.

As the referee pointed out, the column densities of the rows 1 and 2 in Table 1 were H_I column densities, and those of the rows 3 and 4 were H₂ column densities. Since we adopted a formula for computing the dimensionless mass-to-flux ratio λ based on the H₂ column density (this is now stated explicitly in the revised version and subscript H₂ is used the column density N), we have now converted the H_I column densities from Ref. 26 (owing to the changes of references, the Ref. 27 in the previous draft is Ref. 26 now) for the CNMs in the first two rows into equivalent H₂ column densities (for the purposes of computing λ in the last column of Table 1). We thank the referee for

the comments, which made us realize that the mass-to-flux ratios for the CNMs are lower than our original estimates by a factor of 2; this oversight is now corrected. This correction does not change our qualitative conclusions, because the CNMs remain highly magnetically subcritical. Because the column density in row 4 is that for H₂, it is indeed similar to the value estimated by Ref. 17 based on OH abundance (our N_{H_2} of $7 \times 10^{21} \text{ cm}^{-2}$ versus their $9 \times 10^{21} \text{ cm}^{-2}$). We have now pointed out this agreement in one of the notes on Table 1.

We agree that the HiNSA may trace a somewhat different regime from the dust emission. Because the 2.9' resolution of the HiNSA column density map is much larger than that of the dust column density map, the spatial extents of HiNSA and dust emission may differ less than a factor of 3 from the appearance of Fig. 1a. Following the suggestions of the referee, we have explicitly stated that the spatial extents of HiNSA is substantially larger than those of dust emission. We also have added the last sentence in the first paragraph of page 3 to explain our assumption of using the column densities obtained from dust emission as the column densities traced by HiNSA .

In addition, in the third section of Method, we have added a paragraph (lines 297–305) to note the systematic difference between the two methods and explain that the systematic difference should not change the qualitative conclusion of our work.

(2) They state that their result shows that the reduction in magnetic flux relative to mass occurs earlier than envisioned in the “classical” theory of star formation, but the discussion of this is rather limited (lines 141-149). Yet this is THE astrophysical result of the paper. They should

briefly describe the “classical” prediction, namely that mass/flux is reduced by gravity driving neutrals through the ions and magnetic field by a process called ambipolar diffusion. They should explicitly state exactly why their result is contrary to this prediction; that is, what is the argument that the regime sampled by the H_I self-absorption is not gravitationally contracting with ambipolar diffusion producing a subcritical region as in the “classical” theory. Further, they should briefly suggest how their result might be explained theoretically if ambipolar diffusion does not. Two major possibilities are (1) formation of molecular clouds by flows along flux tubes (i.e., Vazquez-Semadeni et al., MNRAS 414, 2511, 2011) and (2) magnetic reconnection (i.e., Lazarian et al., ApJ 757, 154, 2012). In (1), for a relatively small distance along a flux tube, as sampled by a small telescope beam, initially there will be little mass and λ will be measured to be highly subcritical. As mass flows into the region of the cloud (not due to gravity but due to colliding interstellar flows), λ (again as sampled in a small telescope beam) increases. Thus clouds start out as atomic and subcritical and accumulate mass over large distances to become molecular and supercritical as they evolve, becoming self-gravitating at about the same time. In (2), as the amplitude of turbulence as well as the scale of turbulent motions decrease from the envelope to the core of a cloud, the diffusion of the magnetic field is faster in the envelope. As a result, the magnetic flux trapped during the collapse in the envelope is being released faster than the flux trapped in the core, resulting in much weaker fields in envelopes than in cores. Both of these “non-classical” theories would seem capable of explaining the observational result of this paper.

We thank the referee for detailed comments on the interpretation of our observational results, which we largely agree with. We have added the following to pages 6 and 7: “This early reduction

of flux relative to mass is unlikely due to the “classical” scenario where gravity drives neutrals through ions (and the magnetic field tied to them) in a process called ”ambipolar diffusion” because the CNM is not self-gravitating. It could be achieved in a scenario where molecular clouds start out as atomic and subcritical and accumulate mass along field lines via colliding flows to become molecular and supercritical (Ref. 29). It may also be produced by (turbulence-enhanced) magnetic reconnection (Ref. 30).”

We are hesitant to adopt the referee’s argument that the magnetic flux is diffused away faster in the envelope than in the core because the turbulence is stronger in the former, because it is not clear to us the exact nature of the turbulence in the envelope and core and how it enhances reconnection mechanically.

(3) The statement in lines 138-140 that “coherence” of the magnetic fields means the angles between the fields in the envelope and core are the same should be expanded. It will be subject to the same criticism made in reference 22 to the OH Zeeman results of reference 19, that the angles between the line of sight and fields in the two regions may be different.

While the coherent B_{los} suggests that the inclination angles of magnetic fields in the CNM1 and L1544 envelope are likely to be similar, we agree that the inclination angle of the L1544 core magnetic field would be different from that of the coherent field. An assumption of similar inclination angles of the magnetic fields in the L1544 envelope and core thus is required in order to calculate the relative values of λ . We have revised the sentence from lines 112 to 114 to explicitly

state the assumption.

In addition, we have added the last three paragraphs of Method to explain that our assumption might be reliable. Our Monte Carlo simulations suggest that the the three-times difference between the B_{los} of the L1544 envelope and core is less likely to be solely explained by different inclination angles.

(4) In table 2 the authors state the order of the five components along the line of sight, without explaining how they arrived at this order. Of course the self-absorption must be closer than at least one emission component, but how was the specific order determined? In particular, would a different order affect their fitting results?

In our fittings, we fix order = 0 for HiNSA since Taurus cloud is one of the closest cloud to us, and the orders of the other four components are free parameters. The fitted orders of CNM1, CNM2, and CNM3 are in the same sequence as those obtained from the H α absorption toward 3C133 in Ref. 26.

If we fix order = 1 for HiNSA , the order of CNM1 becomes 2, and the order of CNM2 becomes 0. We list the fitted parameters with fixed order = 1 for HiNSA in Table 1, and we show the fitted $I(\nu)$ and $V(\nu)$ spectra in Figure 1. The fitted parameters of I_{peak} , τ , v_{LSR} , σ_v , and B_{los} of the 5 components are similar to those in Extended Data Table 1 of the manuscript. Since the overlapped velocity range of CNM1 and CNM2 is small, CNM1 and CNM2 are almost independent to each

other in the radiative transfer process. It is therefore reasonable that changing HiNSA from order = 0 to order = 1 only changes the orders of CNM1 and CNM2, and the other fitted parameters remain similar. Since the fitting is not sensitive to HiNSA order = 0 or order = 1 (despite that HiNSA order = 0 is more intuitive than HiNSA order = 1), the orders of CNM1 and CNM2 are not the main focus of this work. The test of HiNSA order = 1 shows that the B_{los} of HiNSA and CNM1 are robust, and our conclusion of coherence magnetic field from CNM1 to L1544 would not be changed if HiNSA order = 1.

We further perform a test with fixed HiNSA order = 1 and CNM3 order = 0. The fitted results are shown in Table 2 and Figure 2. The results are less reliable than the two previous cases. The CNM1 in this case has a central velocity $\sim 4 \text{ km s}^{-1}$, inconsistent with the central velocity of CNM1 toward 3C133 in Ref. 26. In addition, the $V(v)$ profiles of CNM1 and CNM3 appear to be fake since the opposite $V(v)$ shapes of the CNM1 and CNM3 compensate each other and generate a flat spectrum of total $V(v)$. Also, the sum of the Zeeman splitting profiles of the five components is worse than those of the two previous cases. The test with fixed HiNSA order = 1 and CNM3 order = 0 indicates that it is unlikely to have CNM3 as the closest component.

Table 1: **Parameters with Fixed HiNSA Order = 1**

Component	I_{peak} [K]	τ	v_{LSR} [km s $^{-1}$]	σ_v [km s $^{-1}$]	B_{los} [μ G]	Order
HiNSA	–	0.32 ± 0.01	6.97 ± 0.01	0.40 ± 0.01	$+3.8 \pm 0.3$	1
CNM1	92.12 ± 4.54	0.94 ± 0.13	8.03 ± 0.12	1.86 ± 0.05	$+4.1 \pm 1.0$	2
CNM2	117.64 ± 1.27	0.92 ± 0.10	-0.03 ± 0.36	2.53 ± 0.14	-6.9 ± 0.9	0
CNM3	137.08 ± 3.00	10.89 ± 1.00	4.41 ± 0.08	1.98 ± 0.06	$+3.1 \pm 0.5$	3
WNM	46.76 ± 2.51	–	2.64 ± 0.04	6.44 ± 0.09	-3.5 ± 1.7	4

Figure 1: The fitted $I(v)$ and $V(v)$ spectra with fixed HiNSA order = 1. Please see the captions of Fig. 2 and 3 in the manuscript.

Table 2: **Parameters with Fixed HiNSA Order = 1 and
CNM3 Order = 0**

Component	I_{peak} [K]	τ	v_{LSR} [km s $^{-1}$]	σ_v [km s $^{-1}$]	B_{los} [μ G]	Order
HiNSA	–	0.32 ± 0.01	6.98 ± 0.01	0.39 ± 0.01	$+3.8 \pm 0.4$	1
CNM1	112.56 ± 0.82	7.07 ± 0.29	3.95 ± 0.01	2.73 ± 0.02	$+1.4 \pm 0.4$	2
CNM2	67.12 ± 14.31	0.59 ± 0.33	-3.22 ± 0.16	1.29 ± 0.04	-5.4 ± 1.7	3
CNM3	128.84 ± 0.12	7.70 ± 0.69	3.20 ± 0.01	1.11 ± 0.03	$+2.6 \pm 0.9$	0
WNM	47.32 ± 1.35	–	2.47 ± 0.03	6.43 ± 0.05	-5.1 ± 1.8	4

Figure 2: The fitted $I(v)$ and $V(v)$ spectra with fixed HiNSA order = 1 and CNM3 order = 0. Please see the captions of Fig. 2 and 3 in the manuscript.

(5) These additions may lead to the paper being too long. If so, I would recommend that the detailed discussion of maximum likelihood and figure 4 be deleted. The statement in lines 106-111 is adequate, and the above points are more important than the detailed discussion.

Because the length of the revised Method section does not exceed the limit of 3000 words, we keep the discussion of maximum likelihood and Extended Data Figure 1 in the manuscript.

(6) In line 58 they state that "...H_I... is less affected by depletion at high densities than heavier molecules...". They should provide a reference for this assertion.

We cite Goldsmith et al. (ApJ, 654, 273, 2007) which uses a time-dependent model with H₂ photodissociation including self-shielding to study the evolution of H_INSA in dense clouds. One of the major findings of the model is that after a few 10⁶ years, the density of H_INSA in 10 K cores would reach a very low steady state with $n \sim 2 \text{ cm}^{-3}$. Since H_INSA could exist in the densest and coldest regime of dense cores, the chemical property of H_INSA is less affected by depletion which makes most heavier molecules inaccessible.

(7) Just to have the paper be very clear, I recommend that they state explicitly the + sign for the magnetic fields rather than just have the reader assume that no sign means +. The direction of the field is crucial in their argument of a "coherence" to the magnetic field in this region.

Done.

(8) *In the sentence beginning on line 86, they assert that the weak field strengths are broadly consistent with the H-band polarization angles. This assertion is so broad and nebulous that I recommend the sentence be removed. Or, I know that Clemens has inferred field strengths from his H-band data to which the authors refer. The authors could ask Clemens for permission to quote his as yet unpublished results if they feel this point is important.*

Since the length of the text is very close to the limit, we have removed the sentence.

(9) *Whenever B is used to mean the total field strength, I recommend adding the subscript “total” to B for clarity; and again for better clarity, in table 1, add a footnote to λ stating that $B_{total} = 2B_{los}$ is used in the calculation.*

Done.

(10) *The authors refer to the fact that the data were obtained with a 19-beam system, but it is unclear what if any use was made of results from other than the central beam. Were they used to produce the H α self-absorption map of fig. 1a or was only the central beam used with multiple pointings? What about Zeeman results for the other 18 beams to produce a magnetic field map?*

The polarization commissioning of the 18 off-center beams is not completed yet. The results represented here were made with only the central beam. We have added the last paragraph of page 16 to explain the usage of the 19-beam data.

Beside the above items, we checked our calculations and found that the B_{los} of the WNM component should be $-3.0 \pm 1.7 \mu\text{G}$, instead of $+3.0 \pm 1.7 \mu\text{G}$ in Table 2 of the previous draft. Since the B_{los} of WNM is not used in the main text, this modification would not change the other parts of the manuscript.

Best Regards,

Tao-Chung Ching

Reviewer Reports on the First Revision:

Referee #1 (Remarks to the Author):

The authors have essentially fully complied with the comments and suggestions I made in my first review, where I recommended that this paper be published after author consideration of my review. My response to the points A-H above remain unchanged so I will not repeat them here. I strongly recommend that this paper be published, for the reasons discussed in my first review.

However, there are two points raised by changes made by the authors in the revised manuscript that I think need attention.

- 1) On page 14 lines 5-6 of their response to my review, that authors said that they had added a note to table 1 pointing out the agreement between the column densities obtained from dust emission and from OH, but they seem to have failed to add the note. I request that they do so.
- 2) In response to my comment, between lines 323 and 348 they have added an extensive discussion of the angle between the magnetic field in the envelope and core regions of L1544. I do have several issues with this new material:
 1. On line 328 they use the word "indicating", which is too strong. I suggest replacing this word with "suggesting the possibility".
 2. Similarly, on line 337 I suggest replacing "is" with "may be".
 3. They state that the field strength ratio between core and envelope is a factor of three, while it is actually $10.8/4.1 = 2.6$, so three exaggerates the number; it would be better simply to use 2.6 rather than three.
 4. In lines 347-348 they state that the above ratio is less likely to be explained by different inclination angles than by something else. There is an obvious physical explanation for this ratio, which I strongly think should be mentioned here. Namely, the OH result for the core is highly likely to be sampling higher volume density gas than the envelope density. This is suggested by the factor of about 2 higher column density, listed in table 1, and is supported by modeling studies of the cloud that they reference. If B scales with volume density with the $2/3$ power, the higher B in the core would be due to a higher volume density by a factor of about 4. This seems to be a completely reasonable explanation, supported by the authors' finding that an angle difference is unlikely to be the explanation.

Referee #2 (Remarks to the Author):

This article provides the observational study of the line-of-sight component of the magnetic field strength in a well-known star forming dense core, L1544, and gives an interesting logic and conclusion that the core is in magnetically supercritical state. Whether star forming dense cores are supported against self-gravitational collapse in the early evolutionary phase or not still remains as an important question in the star formation studies. Nowadays many people in the community of the subject seem to think that in general the star forming dense cores are not supported by magnetic field because of many pieces of indirect evidence. However, the direct observational determination of the magnetic field strength has not been done convincingly so far, except for that of Nakamura et al. (2019) who obtained the same conclusion (supercritical) for a well-known core in Taurus Molecular Cloud. Thus, the reviewer thinks that the paper should be published if the conclusion is really convincing.

The review's major concern is on the geometry of the object. The reviewer understands that HiNSA traces the cold and high density part of the cloud. However, its peak is shifted from the center of L1544 by 0.15pc, which is more than the size of the core ~ 0.1 pc. What is the physical relation between L1544 core and the HiNSA? Do authors have a reasonable explanation why the contours of 70% and 90% of the peak HINSA column density in Figure 1 do not enclose L1544?

This concern is also related to the volume density of the envelope of the core at which Zeeman measurement was conducted. The peak volume density of L1544 was previously estimated to be $\sim 2 \times 10^6/\text{cc}$ as the authors mentioned (cf. Caselli et al. 2019 reported $1 \times 10^7/\text{cc}$). But the volume density probed by the present observation can be roughly estimated to be $N_{\text{H}_2}/L = 7 \times 10^{21} \text{ cm}^{-2} / 0.1 \text{ pc} = 2.3 \times 10^4 (\text{cm}^{-3})$ where $L=0.1 \text{ pc}$ is the size of HiNSA peak in the line of sight. Note that the reviewer intentionally chose small L and larger L may result in even smaller volume density. The reviewer could not find reasonable explanation why such a low density region does not envelop L1544. The authors should point out if the reviewer is missing something.

Comment A

Ref. 11 and 14 have analyzed the observation of this object and obtained the opposite conclusions on the criticality of the magnetic field strength: The former concludes the core is rapidly collapsing and the latter favors magnetically retarded collapse. This issue is not settled down and remains as a controversy in the community as far as the reviewer is aware. Since this paper provides a direct conclusion on this issue, the authors should explicitly state this controversy, at least, briefly.

Comment B

At the very end of the main text the authors wrote "It could be achieved in a scenario where molecular clouds start out as atomic and subcritical and accumulate mass along field lines via colliding flows to become molecular and supercritical 29." The authors seem to be considering that both the molecular cloud formation and core formation are occurring in a single event as studied in Ref. 29, which is very different from the current picture of molecular cloud formation (see, e.g., review in Ref. 2). Indeed, the results of calculations shown in Ref. 29 are very crowded distribution of many cores and significantly different from the observed picture of Taurus Molecular Cloud. The reviewer thinks that the essential suggestion from the present article is that the core should be created as a magnetically supercritical core, but the mechanism remains to be studied. Thus, rewarding is recommended.

Minor Comment:

Caption of Figure 3: WMN components ==> WNM components

Ref.)

"The Central 1000 au of a Pre-stellar Core Revealed with ALMA. I. 1.3 mm Continuum Observations", Caselli et al. (2019) ApJ 874, 89

"First clear detection of the CCS Zeeman splitting toward the pre-stellar core, Taurus Molecular Cloud 1", Nakamura et al. (2019) PASJ, 71, 117

Author Rebuttals to First Revision:

Thank you very much for revising our manuscript. We appreciate the careful reviews and constructive suggestions, and we have taken all the comments into account. Here we are resubmitting our paper with revisions based on the comments from you and from the editor. The changes in this second revision are highlighted in blue color in the manuscript, while the changes in the first revision are printed in black color. Below we list our answers to the comments.

Referee 1's Comments:

(1) On page 14 lines 5-6 of their response to my review, that authors said that they had added a note to table 1 pointing out the agreement between the column densities obtained from dust emission and from OH, but they seem to have failed to add the note. I request that they do so.

REPLY: We are sorry that in our reply in June, we forget to add the note to Table 1. It is now added on lines 210–211.

(2) In response to my comment, between lines 323 and 348 they have added an extensive discussion of the angle between the magnetic field in the envelope and core regions of L1544. I do have several issues with this new material: 1. On line 328 they use the word “indicating”, which is too strong. I suggest replacing this word with “suggesting the possibility”.

REPLY: We have rephrased the discussion in this paragraph, and we stick to a more modest

tone throughout the discussion, following the referee's comment.

(3) *Similarly, on line 337 I suggest replacing "is" with "may be".*

REPLY: Done, on line 357.

(4) *They state that the field strength ratio between core and envelope is a factor of three, while it is actually $10.8/4.1 = 2.6$, so three exaggerates the number; it would be better simply to use 2.6 rather than three.*

REPLY: Done

(5) *In lines 347-348 they state that the above ratio is less likely to be explained by different inclination angles than by something else. There is an obvious physical explanation for this ratio, which I strongly think should be mentioned here. Namely, the OH result for the core is highly likely to be sampling higher volume density gas than the envelope density. This is suggested by the factor of about 2 higher column density, listed in table 1, and is supported by modeling studies of the cloud that they reference. If B scales with volume density with the $2/3$ power, the higher B in the core would be due to a higher volume density by a factor of about 4. This seems to be a completely reasonable explanation, supported by the authors' finding that an angle difference is unlikely to be the explanation.*

REPLY: We thank referee for the instructional suggestion. We have rephrased the discussion

on lines 343–348 to include the physical explanation of increasing magnetic field strength with increasing density.

Referee 2's Comments:

(1) However, the direct observational determination of the magnetic field strength has not been done convincingly so far, except for that of Nakamura et al. (2019) who obtained the same conclusion (supercritical) for a well-known core in Taurus Molecular Cloud. Thus, the reviewer thinks that the paper should be published if the conclusion is really convincing.

REPLY: CCS probes a parameter regime that is different from that probed by HiNSA . Nakamura et al. noted that the region that CCS probed has an H_2 column density of $3 \times 10^{22} \text{ cm}^{-2}$, which is higher than that probed by the OH in L1544 ($\sim 7 \times 10^{21} \text{ cm}^{-2}$) and nearly one order of magnitude higher than that probed by HiNSA ($\sim 3.5 \times 10^{21} \text{ cm}^{-2}$). HiNSA probes a wide range of densities, due to its robust and simple chemistry, and a steady state produced by a balance between H_2 formation and destruction, which result in HiNSA strength being largely independent of gas densities (Ref. 5).

For established systematic Zeeman probes, namely, HI, OH, and CN, there is a density gap between HI and OH, which exactly covers the expected critical break-point density (Ref. 15). The characteristics of HiNSA described above and the successful Zeeman measurement reported here thus provide crucial information about the transition from magnetically subcritical to supercritical state.

We have added the citation to the CCS detection and related discussions to the first paragraph of the main text. More detailed discussion of the CCS Zeeman measurement was added to the Method as follows

”Ref. 20 reported a CCS Zeeman detection of $117 \pm 21 \mu\text{G}$ in a dense core of TMC-1 that has an estimated H_2 column density of $3 \times 10^{22} \text{ cm}^{-2}$, which is 4 times higher than that probed by OH Zeeman measurements in L1544 and nearly one order of magnitude higher than that probed by our HiNSA measurements. It appears to provide further support to the evolutionary scenario suggested by our HiNSA measurements: namely, once the gas loses its magnetic support during the transition from CNM to the molecular envelope (or ridge) and becomes magnetically supercritical, there is no longer any need to lose magnetic flux further (relative to the mass) in order for a piece of the envelope/ridge to condense into a (magnetically supercritical) core (e.g., the L1544 core probed by OH) and for the core to evolve further by increasing its column density (e.g., the TMC-1 core probed by CCS).

Technically, we note that one potential source of significant uncertainty in frequency shift, namely the uncertainty of beam squint, was not included in the CCS result, which may affect the level of significance. In comparison, the HiNSA measurement is robust with a $> 10\sigma$ significance with the beam squint and velocity gradient been taken into account by convolving the FAST Stokes V beam with the Stokes I cube of L1544 (see the third paragraph in the section of data reduction in Methods)”

(2) The review’s major concern is on the geometry of the object. The reviewer understands that

HiNSA traces the cold and high density part of the cloud. However, its peak is shifted from the center of L1544 by 0.15 pc, which is more than the size of the core ~ 0.1 pc. What is the physical relation between L1544 core and the HiNSA? Do authors have a reasonable explanation why the contours of 70% and 90% of the peak HiNSA column density in Figure 1 do not enclose L1544?

REPLY: We first note that dense gas tracers, such as CCS, has also been seen to offset from the dust (e.g., Ref. 41). Such an offset can result from complexities in chemistry and formation history, but does not affect the main science result of this work, namely, HINSA Zeeman probes previously uncovered density gap around the critical break- point density ($\sim 300 \text{ cm}^{-3}$). Regardless of the details of further evolution to higher densities, the magnetically supercritical state seems to have been achieved in the molecular envelope which has a density close to the break-point density.

We discuss here in more details, the offset seen in L1544, which is arguably the most intensively studied prestellar core on the brink of star formation. Its geometry may not be as simple as envisioned in idealized theories, where the dense core sits near the center of a lower density molecular envelope. In particular, the L1544 core appears to sit near one end of an elongated molecular (and dust) ridge, which roughly coincides with the region traced by HiNSA (see Figure 1a in the manuscript). The exact reason for the offset of the core from the geometric center of the envelope is unclear. It could be due to the so-called “gravitational edge focusing” effect discussed in, e.g., F. Heitsch et al. (2009, ApJ, 704, 1735), where gravitational collapse tends to occur near the end of a cylinder (or the edge of a flattened sheet). It could also be due to external compression of the system from one side, as indicated by the steeper column density gradient on the lower side of the

core compared to the upper side; the compression could have facilitated the core condensation in the first place. Although it is unclear how exactly the dense core forms, we believe it is reasonable to assume that it is condensed out of the molecular ridge that includes both the material that is currently residing in the core and the remaining (or remnant) ridge material that was not incorporated into the core and that is currently traced by H_INSA. In other words, in our picture, the current molecular ridge traced by H_INSA is envisioned to be similar to the original ridge material that forms the dense core (i.e., the ridge is the progenitor of the core). It is in this sense that we refer to the H_INSA ridge as “the molecular envelope” of the core despite the apparent offset between the core and the peak of the H_INSA absorption. The discussion above has been added to Methods on lines 280–282 and 312–320 to make this point clear.

(3) This concern is also related to the volume density of the envelope of the core at which Zeeman measurement was conducted. The peak volume density of L1544 was previously estimated to be $\sim 2 \times 10^6/\text{cc}$ as the authors mentioned (cf. Caselli et al. 2019 reported $1 \times 10^7/\text{cc}$). But the volume density probed by the present observation can be roughly estimated to be $N_{\text{H}_2}/L = 7 \times 10^{21} \text{ cm}^{-2} / 0.1\text{pc} = 2.3 \times 10^4 \text{ (cm}^{-3}\text{)}$ where $L=0.1\text{pc}$ is the size of H_INSA peak in the line of sight. Note that the reviewer intentionally chose small L and larger L may result in even smaller volume density. The reviewer could not find reasonable explanation why such a low density region does not envelop L1544. The authors should point out if the reviewer is missing something.

REPLY: We agree with the estimate above and would like to refer to our reply to comments (2). In short, H_INSA strength is expected to be largely independent of total gas density, due to the

H₂ formation rate and destruction both scale with gas density. Given the beam size of the current radio data, the average gas density traced by H_INSA is around the critical break-point density, which, in the case of L1544, is located in the molecular envelope encompassing the dense core.

(4) Comment A: Ref. 11 and 14 have analyzed the observation of this object and obtained the opposite conclusions on the criticality of the magnetic field strength: The former concludes the core is rapidly collapsing and the latter favors magnetically retarded collapse. This issue is not settled down and remains as a controversy in the community as far as the reviewer is aware. Since this paper provides a direct conclusion on this issue, the authors should explicitly state this controversy, at least, briefly.

REPLY: Thanks for the excellent suggestion! We have added a sentence on lines 123–124 to state our preference for Ref. 8 (was Ref. 11) over Ref. 9 (was Ref. 14) because the magnetic field in the envelope does not appear to be strong enough to retard the collapse significantly.

(5) Comment B: At the very end of the main text the authors wrote “It could be achieved in a scenario where molecular clouds start out as atomic and subcritical and accumulate mass along field lines via colliding flows to become molecular and supercritical²⁹.” The authors seem to be considering that both the molecular cloud formation and core formation are occurring in a single event as studied in Ref. 29, which is very different from the current picture of molecular cloud formation (see, e.g., review in Ref. 2). Indeed, the results of calculations shown in Ref. 29 are very crowded distribution of many cores and significantly different from the observed picture of Taurus

Molecular Cloud. The reviewer thinks that the essential suggestion from the present article is that the core should be created as a magnetically supercritical core, but the mechanism remains to be studied. Thus, rewarding is recommended.

REPLY: We have added the caveat on lines 131–133 that “whether these scenarios can reproduce the distributions of gas and magnetic field observed in the L1544 region quantitatively remains to be seen.”

(6) *Minor Comment: Caption of Figure 3: WMN components => WNM components*

REPLY: Done

(7) *Ref.*

“The Central 1000 au of a Pre-stellar Core Revealed with ALMA. I. 1.3 mm Continuum Observations”, Caselli et al. (2019) ApJ 874, 89

“First clear detection of the CCS Zeeman splitting toward the pre-stellar core, Taurus Molecular Cloud I”, Nakamura et al. (2019) PASJ, 71, 117

REPLY: Done

Best Regards,

Tao-Chung Ching and Di Li

Reviewer Reports on the Second Revision:

Referee #1 (Remarks to the Author):

This paper presents an important new observational result on the importance of magnetic fields during the evolution of interstellar clouds through the diffuse and molecular cloud stages to star formation. I will not repeat here my detailed comments made in previous reviews. You have satisfied all reviewer concerns, and due to the unique nature of these important new results I have recommended that the paper be published in its present form.

Referee #2 (Remarks to the Author):

The authors' response is reasonable for most of my points raised in the previous report. However, I still have one concern in the point (5) Comment B.

>> (5) Comment B

At the very end of the main text the authors wrote "It could be achieved in a scenario where molecular clouds start out as atomic and subcritical and accumulate mass along field lines via colliding flows to become molecular and supercritical 29." The authors seem to be considering that both the molecular cloud formation and core formation are occurring in a single event as studied in Ref. 29, which is very different from the current picture of molecular cloud formation (see, e.g., review in Ref. 2). Indeed, the results of calculations shown in Ref. 29 are very crowded distribution of many cores and significantly different from the observed picture of Taurus Molecular Cloud. The reviewer thinks that the essential suggestion from the present article is that the core should be created as a magnetically supercritical core, but the mechanism remains to be studied. Thus, rewarding is recommended.

> REPLY: We have added the caveat on lines 131–133 that "whether these scenarios can reproduce the distributions of gas and magnetic field observed in the L1544 region quantitatively remains to be seen."

I think the newly inserted sentence still means that the possible scenarios are those two, which is not acceptable to me and possibly to many theorists. The reason is that the first scenario (colliding flows) shown in Ref. 29 does not show the structure that resembles Taurus molecular cloud in any sense. (Remember that the authors of Ref. 29 pushed the molecular material along the magnetic field lines to create super-critical dense gas.) The second scenario requires significantly strong turbulence that does not seem to be consistent with the narrow line-widths of molecular emission lines (e.g., CCS, N₂H⁺) of L1544. I think it's better to write something like "this observation provides a new specific question on how to create super-critical dense cores such as L1544 from sub-critical molecular clouds". If the authors want to keep the two scenario as possible explanations, I would strongly recommend to remove the word "quantitatively" in the newly inserted sentence.

[Minor Comment]

In an added sentence in Page 2 (Line 32), the authors wrote "critical density" without defining its actual meaning. A brief explanation is recommended.

Author Rebuttals to Second Revision:

We thank the referees for reviewing our manuscripts and providing useful comments. This is a point-by-point response to the remaining issues raised by the referees. Below we list our answers to the Referees' comments.

Referee 1

(1) This paper presents an important new observational result on the importance of magnetic fields during the evolution of interstellar clouds through the diffuse and molecular cloud stages to star formation. I will not repeat here my detailed comments made in previous reviews. You have satisfied all reviewer concerns, and due to the unique nature of these important new results I have recommended that the paper be published in its present form.

REPLY: We thank Referee 1's recommendation on our manuscript.

Referee 2

(1) The authors' response is reasonable for most of my points raised in the previous report. However, I still have one concern in the point (5) Comment B.

>> (5) Comment B

At the very end of the main text the authors wrote "It could be achieved in a scenario where molecular clouds start out as atomic and subcritical and accumulate mass along field lines via colliding flows to become molecular and supercritical 29." The authors seem to be considering that both

the molecular cloud formation and core formation are occurring in a single event as studied in Ref. 29, which is very different from the current picture of molecular cloud formation (see, e.g., review in Ref. 2). Indeed, the results of calculations shown in Ref. 29 are very crowded distribution of many cores and significantly different from the observed picture of Taurus Molecular Cloud. The reviewer thinks that the essential suggestion from the present article is that the core should be created as a magnetically supercritical core, but the mechanism remains to be studied. Thus, rewarding is recommended.

> REPLY: We have added the caveat on lines 131–133 that “whether these scenarios can reproduce the distributions of gas and magnetic field observed in the L1544 region quantitatively remains to be seen.”

I think the newly inserted sentence still means that the possible scenarios are those two, which is not acceptable to me and possibly to many theorists. The reason is that the first scenario (colliding flows) shown in Ref. 29 does not show the structure that resembles Taurus molecular cloud in any sense. (Remember that the authors of Ref. 29 pushed the molecular material along the magnetic field lines to create super-critical dense gas.) The second scenario requires significantly strong turbulence that does not seem to be consistent with the narrow line-widths of molecular emission lines (e.g., CCS, N₂H⁺) of L1544. I think it’s better to write something like “this observation provides a new specific question on how to create super-critical dense cores such as L1544 from sub-critical molecular clouds”. If the authors want to keep the two scenario as possible explanations, I would strongly recommend to remove the word “quantitatively” in the

newly inserted sentence.

REPLY: Done. The coherent magnetic fields reviewed here suggest an early transition to supercritical state before what is expected in the classical model of ambipolar diffusion. It would have happened in the scenarios of Ref. 29 and 30, with a caveat that the cloud structure of in Ref. 29 does not resembles the observed molecular clouds. Since Referee 1 requested to cite Ref. 29 and 30, we would like to keep the two citations. In the mean time, we have followed Referee 2's suggestion to remove the word "quantitatively" and refresh the statement as the lines 130–133 in the manuscript.

(2) [*Minor Comment*]

In an added sentence in Page 2 (Line 32), the authors wrote "critical density" without defining its actual meaning. A brief explanation is recommended.

REPLY: Done. We have added a brief explanation of the critical density on lines 32–33 in the manuscript.

Best Regards,

Tao-Chung Ching and Di Li